# Effect of Silicon Content on Microstructures and Properties of Directionally Solidified Fe-B Alloy

**DOI:** 10.3390/ma15175937

**Published:** 2022-08-27

**Authors:** Pengjia Guo, Shengqiang Ma, Xuebin He, Intizar Ali Shah, Ping Lv, Hantao Chen, Jiandong Xing, Liujie Xu, Jiankang Zhang

**Affiliations:** 1National Joint Engineering Research Center for Abrasion Control and Molding of Metal Materials, Henan University of Science and Technology, Luoyang 471000, China; 2State Key Laboratory for Mechanical Behavior of Materials, School of Materials Science and Engineering, Xi’an Jiaotong University, Xi’an 710049, China; 3Shaanxi Union Research Center of University and Enterprise for Zinc-Based New Materials, Xi’an 710049, China

**Keywords:** directionally solidified Fe-B alloy, microstructure, solid solution strengthening, silicon

## Abstract

In order to investigate the effect of Si content on the microstructures and properties of directionally solidified (DS) Fe-B alloy, a scanning electron microscope (SEM) with an energy dispersive spectrum (EDS), and X-ray diffraction have been employed to investigate the as-cast microstructures of DS Fe-B alloy. The results show that Si can strongly refine the columnar microstructures of the DS Fe-B alloy, and the columnar grain thickness of the oriented Fe_2_B is reduced with the increase of Si addition. In addition, Si is mainly distributed in the ferrite matrix, almost does not dissolve in boride, and seems to segregate in the center of the columnar ferrite to cause a strong solid solution strengthening and refinement effect on the matrix, thus raising the microhardness of the matrix and bulk hardness of the DS Fe-B alloy.

## 1. Introduction

In many high-temperature manufacturing fields, the rollers and tools acting as forming or molding equipment are of importance in the production of parts and components. However, the corrosion and wear of mechanical equipment can cause great economic losses. Therefore, in order to reduce these losses, the research of many excellent wear-resistant and corrosion-resistant materials has been given high priority in the field of materials science in recent years [1,2,3]. It is very important to develop new materials with low cost and outstanding wear resistance for industrial applications [4]. High-speed steel, such as M2 high-speed steel, has excellent hardness, wear resistance, and high temperature performance, and is widely used in high-speed machining and cutting operations [5,6,7,8]. However, high-speed steel rolls contain a large amount of expensive alloying elements such as tungsten, molybdenum, cobalt, vanadium, and niobium, which increases the production cost and causes the application to be greatly limited [5,9,10].

Currently, boron is introduced into alloys as a new alloying method to form hard phases in order to develop a new type of wear-resistant and fatigue-resistant high-speed steel roll [11,12,13,14]. The Fe-B alloy possesses high hardness and wear resistance owing to the alloyed borides with improved performances [15]. Nowdays, the improvement of Fe-B alloy in wear resistance mainly depends on the unique microstructure of Fe-B-C alloy as a new tool steel, and its property can be effectively and well controlled by B and C contents to obtain suitable and desired properties [16,17,18,19,20]. In addition, the researchers found that adding boron can also improve the wear resistance of other alloys such as ductile iron, high-chromium cast iron, and high-strength steels [4,21,22,23,24,25].

At present, the role of Si in high-speed steel has attracted more attention. Pan et al., found that with the increase of Si, the content of M_2_C carbides in the microstructures of the steels decreased sharply, and the amount of M_6_C carbides increased rapidly [26]. The addition of only small quantities of Si can alter the microstructure and carbide types in steels, leading to significant enhancement in the mechanical properties of the material [27]. Bhadeshia and Cai et al. studied the role of Si in silicon steel and austenitic stainless steel, which indicates that Si can promote segregation of other alloying elements, which in turn increases material strength [28,29]. From the discussions mentioned above, it can be seen that Si plays a crucial role in determining the microstructures and overall properties of materials. Furthermore, the orientation and size of the hard phases in cast irons can strongly determine the mechanical and wear-reisitant properties of the materials, which may depend on the alloying elements and solidification rates to refine the microstructures [30,31]. Liu et al. investigated the effect of Si on the corrosion resistance of DS Fe-B alloy in static liquid zinc, and found that Si improves the corrosion resistance to liquid zinc [32]. Prince Setia et al. found that ~4 wt.% Si addition into stainless steel can give rise to phase transformation from a single-phase austennite to duplex microstructure of ferrite and austenite owing to the ferrite stabilizer effect of Si, e.g., the increase in the activity of carbon and solid solution strengthening of the Si addition [27].

However, there is little research on the influence of Si addition on the microstructure and properties in the directionally solidified (DS) Fe-B alloy. Additionally, the influence mechanism and refinement of Si on the DS Fe-B alloy is still unclear. Therefore, in the present work, the as-cast DS Fe-B alloy with various Si contents is investigated and discussed to further reveal the effect of Si on the structures and properties of DS Fe-B alloy.

## 2. Materials and Methods

### 2.1. Sample Preparation

The chemical composition of the investigated DS Fe-B alloy containing Si is listed in Table 1. The four samples with different Si contents were denoted as A1, A2, A3, and A4 samples of DS Fe-B alloy. The Fe-B alloy were prepared in a 10-kg intermediate frequency induction melting furnace (Xi’an Yinhai Electric Furance Co., Ltd., Xi’an, China). Firstly, the pure iron and pig iron were melted, and then, ferrochromium and ferrosilicon were added into the furnace in sequence. When all the alloys were melted in the furnace, the preheated raw ferroboron was added after deoxidizing it with a little pure aluminum. Once the alloy was melted at 1450–1480 °C, it was poured into the specially designed mold to solidify in one-way heat dissipation (i.e., opposite to the direction of crystal growth for the Fe-B alloy), obtaining some Y-block ingots of the DS Fe-B alloy, as shown in Figure 1a. The directional region near the chilled copper mold with the cooling of circulating water was selected for analysis, where the sample with a solidification rate of approximately 15 °C/s was measured by a thermocouple [33]. The various morphologies of the oriented Fe_2_B (e.g., black areas) and α-Fe (e.g., white areas) in the longitudinal and transverse sections of the DS Fe-B alloy are schematically illustrated in Figure 1b,c (i.e., two-phase microstructure of α-Fe and Fe_2_B in the DS Fe-B alloy), which indicates that the (002) orientation of Fe_2_B along the longitudinal section is parallel to the growth direction of Fe_2_B crystal, whereas the transverse section of the columanr Fe_2_B (i.e., Fe_2_B (002) crystal plane) is perpendicular to the Fe_2_B growth direction (i.e., opposite to the heat dissipation) [34,35].

### 2.2. Characterization

The cast samples of the DS Fe-B alloy with the dimension of 15 × 10 × 10 mm^3^ were cut from ingots by wire electrode cutting machine. All the tested samples were ground and polished, and then etched by 4 vol% (volume ratio) nitrate alcohol solution to observe the microstructures. The as-cast microstructures were observed by scanning electron microscopy (SEM, VEGAII, XMUINCA, TESCAN, Brno, Czech Republic) with an energy dispersive spectrum (EDS), and an X-ray diffraction (XRD, D/Max-2400X, Rigaku Corporation, Tokyo, Japan). The XRD was directly performed on the as-cast specimens using Cu-Kα radiation coupled with continuous scanning at 40 kV and 200 mA as an X-ray source. The specimens for XRD were scanned in the angle 2θ, ranging from 20° to 100°, with a step size of 0.02° and a collection time of 10 s. The Image-pro plus software (Image-Pro Plus 6.0, Media Cybernetics, Maryland, USA) was used to measure the average thickness of the columnar borides (i.e., d_Fe2B_) from the one edge to another. The bulk hardness of as-cast structure was measured on an HR-150A Rockwell hardness tester (Beijing Shidai Shangfeng Technology Co., Ltd., Beijing, China). The microhardness of boride and matrix in the DS Fe-B alloy was measured by using an HXD-type 1000 Vickers-hardness tester with a load of 100 gf.

## 3. Results and Discussion

### 3.1. As-Cast Microstructure of DS Fe-B Alloy with Various Si Additions

Figure 2 shows the morphologies of the as-cast DS Fe-B alloy with different Si contents. From Figure 2a–d, it can be clearly seen that the microstructures of the four Si-containing specimens show good orientation effects, and it is apparent that microstructures consist of the oriented Fe_2_B phase with a typical faceted crystal growth as an intermetallic compound (e.g., where the growth direction of columnar boride opposite to the heat dissipation direction under the directional solidification condition) and α-Fe matrix, i.e., a typical two-phase microstructure with a ductile phase and a hard phase [34,35]. The columnar gray Fe_2_B borides are arranged in long rods, where the morphology of the oriented Fe_2_B is greatly displayed as straight and tall. In addition, the gray-white α-Fe matrix as the continuous toughening or ductile phase is distributed among the interlayer of the rod-like Fe_2_B, which reveals the formation of the dual-phase oriented microstructures of the DS Fe-B alloys. Especially, the exceeding regular laminated structure of Fe_2_B and α-Fe in DS Fe-B alloy are greatly refined and dense with the increase of Si content [27], and the forked growth phenomenon of oriented Fe_2_B intermetallic compound is reduced with the addition of Si, which infers that Si may promote the strong directional growth of Fe_2_B.

Figure 3 shows the relationship between average thickness of the columnar Fe_2_B grain (d_Fe2B_) and Si contents. From Figure 3a, as schematically defined with the double-headed arrow in it, the average thickness of the columnar borides d_Fe2B_ decreases rapildly when the variation in Si content in Fe-B alloy increases from 0.00 wt.% to 2.50 wt.%, and decreases slowly after the Si content exceeds 2.50 wt.%. It can be seen that with the increase of Si content, the oriented eutectic boride becomes more thinner and smaller, which suggests that Si has a dramatically refining effect on columnar hard-phase Fe_2_B in the DS Fe-B alloy [27,28].

### 3.2. XRD Analysis and Hardness Tests of As-Cast DS Fe-B Alloy

Figure 4 shows the XRD patterns of the as-cast DS Fe-B alloy with different Si contents. It can be seen that the as-cast microstructures of the longitudinal section of DS Fe-B alloy are mainly composed of Fe_2_B (36-1332), α-Fe (06-0696) and/or some Fe-Cr (34-0396) solution [34,35,36], whereas the peaks of the α-Fe matrix gradually move to the right when the Si content exceeds 1.5 wt.% in the DS Fe-B alloy (Figure 4b). This indicates that the ferrite matrix in the as-cast DS Fe-B is gradually transformed to a solid solution of ferrite, owing to the dissolution of the small atomic radius Si into the matrix. Obviously, on one hand, Si may promote the solid solution of Cr atoms in the matrix, resulting in more (Fe-Cr) substitutional solid solutions, and on the other hand, Si is likely in favour of forming the depletion or segregation at the ferrite/boride boundaries to refine the columnar structure of the DS Fe-B alloy, which may be attribute to the constitutional supercooling effect and the outcome of Cr substitutional solid solution caused by the role of Si [27,28,29]. Actually, phase boundary segregation and matrix solution of Si may also adjust the Cr in the microstructures of Fe_2_B and α-Fe to promote the replacement of Fe atoms by Cr atoms in borides, likely forming some Cr-rich borides, as the changes of the boride peaks in XRD [37]. Obviously, the detected peaks of the transverse section in the DS Fe-B alloy comprise the only (002) crystal plane of the Fe_2_B and very strong (110) crystal plane of the α-Fe and both of these peaks are relatively strong (Figure 4c), whereas more peaks of Fe_2_B in the longitudial section appear (Figure 4a).

Figure 5 shows the bulk hardness of the as-cast DS Fe-B alloy and the micro hardness of oriented Fe_2_B in transverse and longitudinal sections. From Figure 5a, it is clear that the bulk hardness of the transverse and longitudinal sections showed a continually upward trend when the Si content increases from 0.00 wt.% to 3.50 wt.%. Meanwhile, the macrohardness of the transverse section in the DS Fe-B alloy is higher than that of the longitudinal section. It means that Si addition can refine the sizes of the oriented Fe_2_B hard phase and matrix, and also strongly promote the facet growth of Fe_2_B (002) crystal plane and refinement owing to the Si segregation, which increases of the bulk hardness [27,28,29]. The microhardness of the Fe_2_B hard phase in the longitudinal and transverse section is shown in Figure 5b, the microhardness of the oriented Fe_2_B in the transverse and longitudinal section shows small fluctuations, and the microhardness of the hard phase of Fe_2_B in the transverse section has higher values than that in the longitudinal section, which actually indicates that the higher bonding energies of Fe-Fe and Fe-B covalent bonds in Fe_2_B crystal and Si addition may reduce the crystal defects and vacancies in Fe_2_B crystal to improve its microhardness. Obviously, the (002) crystal plane of the Fe_2_B hard phase can withstand larger loads and stain, which plays a main role in wear resistance in the transverse DS Fe-B alloy [5].

### 3.3. Element Distribution of As-Cast DS Fe-B Alloy

Figure 6 and Figure 7 shows the EDS mapping of A1 and A3 DS Fe-B alloy. As shown in Figure 6b,c and Figure 7b,c, it can be observed that the distribution of Fe and Cr in the as-cast structure is uniform, whereas the Si and B is concentrated and obviously segregated. Clearly, the Si mainly exists in the matrix whereas it does not appear in the Fe_2_B phase, as shown in Figure 6 and Figure 7. However, the boron can strongly segregate and enrich in the boride to form the Fe_2_B hard phase during directional solidification process. Combined with the line scanning analysis of Figure 8, it can be clearly known that Si is mainly distributed in the ferrite matrix, and almost insoluble in boride. In addition, there is a gradient distribution of Si at the Fe/Fe_2_B interface. Meanwhile, the distribution of Cr in ferrite and Fe_2_B hard phase is relatively uniform, and more concentration of Cr exists in the boride. Figure 9 shows the compositional profiles of Si in matrix between neighboring borides (i.e., the concentration of Si in the matrix), and it can be seen that the distribution of Si follows a parabolic law, and it has the highest Si content in the middle of the ferrite matrix as a peak value, and a little Si content at the phase boundary of the ferrite and borides. As the Si content increases, more Si is enriched and distributed in the ferrite. This also further illustrates the strong solid solution effect of Si in the ferrite matrix and refinement on the dendrite metal matrix [29], which also further refine the interdendrite eutectic structure during the DS process largely beause of the constitutional supercooling and inhibiting growth of Si and Cr on borides in some crystal orientations (e.g., inhibiting growth on the possible crystal orientation of Fe_2_B virtical to the (002) orientation).

## 4. Conclusions

In present study, as-cast DS Fe-B alloy with various Si contents were designed and fabricated. The as-cast DS Fe-B alloy shows good orientation effects. The main conclusions can be summarized as follows:(1).The Si-containing structures of the four components all show good orientation effects, and that the DS Fe-B alloy consists of a columnar Fe_2_B boride and α-Fe ferrite. The columnar Fe_2_B borides display a preferential growth along the (002) orientation of Fe_2_B, which makes the DS Fe-B alloy a dual-phase oriented microstructures.(2).The columnar grain thickness of oriented Fe_2_B decreases sharply with the increase of Si content, which is attributed to the segregation and refinement Si in the DS Fe-B alloy.(3).The bulk hardness of the transverse and longitudinal section showed an upward trend with the increase of Si content, whereas the microhardness of the oriented Fe_2_B in the transverse section has little change, first increasing then decreasing in the longitudinal section.(4).The Si is mainly distributed in the ferrite matrix, and almost does not dissolve in boride. As the Si content increases, more Si is segregated in the center of the columnar ferrite distributed between the two columnar Fe_2_B borides, which results in the refinement of the Fe_2_B hard phase and solid solution strengthening of the ferrite matrix.

## Figures and Tables

**Figure 1 materials-15-05937-f001:**
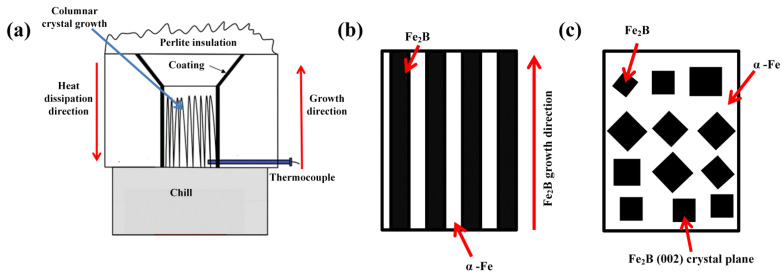
The schematic maps of directional solidification device and orientation plane of DS Fe-B alloy: (**a**) schematic device of DS (directionally solidified) cast Fe-B alloy (solidification rate of about 15 °C/s measured by a thermocouple), (**b**) the schematical morphology of the longitudinal section in DS Fe-B alloy, (**c**) the schematical morphology of the transverse section in DS Fe-B alloy.

**Figure 2 materials-15-05937-f002:**
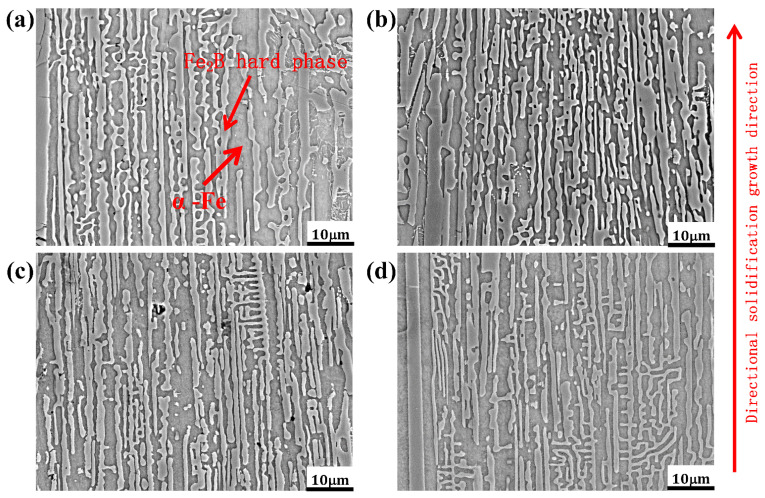
SE morphologies of as-cast DS Fe-B alloy: (**a**) A1, (**b**) A2, (**c**) A3, (**d**) A4.

**Figure 3 materials-15-05937-f003:**
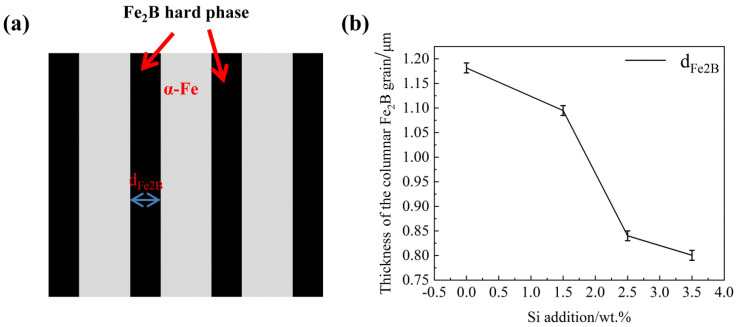
Relationship between average thickness of the columnar Fe_2_B grain (d_Fe2B_) with Si contents: (**a**) the schematic diagram of d_Fe2B_, (**b**) d_Fe2B_ vs. Si content.

**Figure 4 materials-15-05937-f004:**
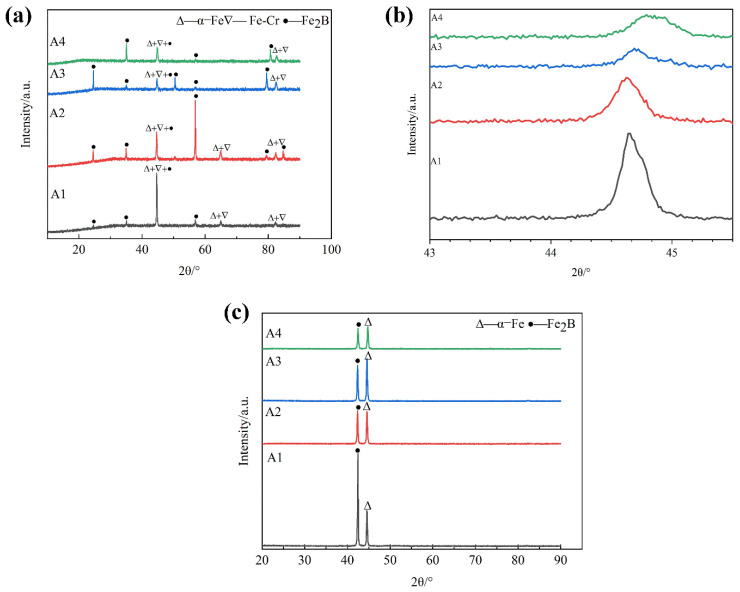
XRD patterns of the as-cast DS Fe-B alloy: (**a**) longitudinal section, (**b**) local peaks of Fe (110) in longitudinal section, (**c**) transverse section.

**Figure 5 materials-15-05937-f005:**
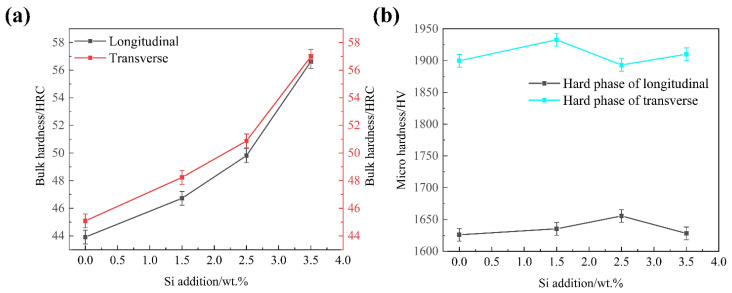
Bulk hardness and micro hardness of as-cast DS Fe-B alloy: (**a**) bulk hardness of longitudinal section and transverse sections, (**b**) microhardness of hard phase.

**Figure 6 materials-15-05937-f006:**
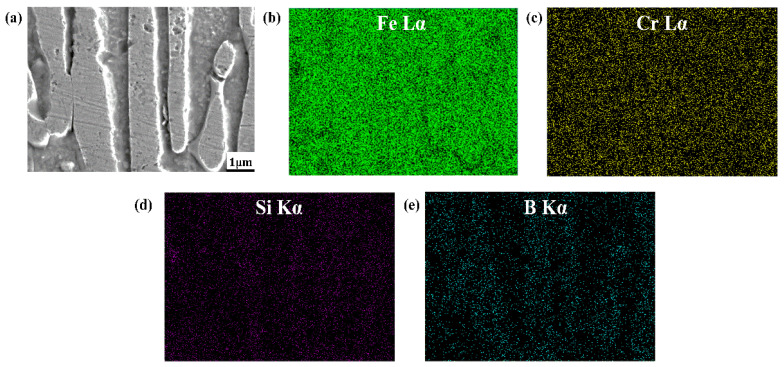
The EDS maps of A1 DS Fe-B alloy: (**a**) micrograph of surface scanning, (**b**) element distribution of Fe, (**c**) element distribution of Cr, (**d**) element distribution of Si, (**e**) element distribution of B.

**Figure 7 materials-15-05937-f007:**
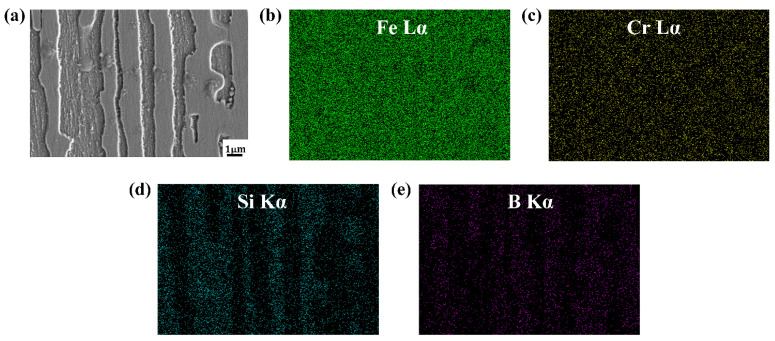
The EDS maps of A3 DS Fe-B alloy: (**a**) micrograph of surface scanning, (**b**) element distribution of Fe, (**c**) element distribution of Cr, (**d**) element distribution of Si, (**e**) element distribution of B.

**Figure 8 materials-15-05937-f008:**
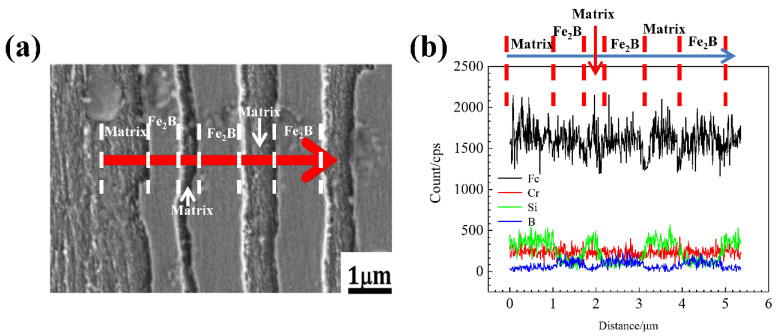
Line scanning analysis of A3 DS Fe-B alloy: (**a**) direction of line scanning, (**b**) peak change of Fe, Cr, Si, and B.

**Figure 9 materials-15-05937-f009:**
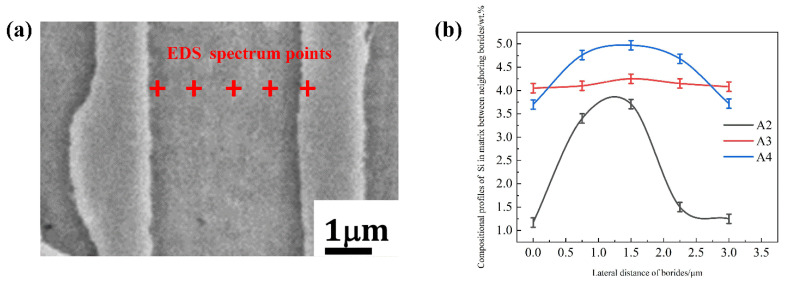
Point scanning analysis of as-cast DS Fe-B alloy: (**a**) schematic map of the Si point scanning of A3 specimen, (**b**) compositional profiles of Si in matrix between neighboring borides.

**Table 1 materials-15-05937-t001:** Chemical compositions of cast DS Fe-B alloy by spark emission spectrometer (wt.%).

Samples	B	Cr	Si	C	Fe
A1	3.51	0.52	0.00	0.11	Bal
A2	3.51	0.50	1.50	0.10	Bal
A3	3.49	0.50	2.50	0.12	Bal
A4	3.50	0.51	3.50	0.11	Bal

## Data Availability

Data sharing is not applicable to this article.

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
