# Peer review of "Effect of Silicon Content on Microstructures and Properties of Directionally Solidified Fe-B Alloy"

_materials, 2022, doi:10.3390/ma15175937_

Round 1

Reviewer 1 Report

introduction need to be complate

Reviewer 2 Report

The authors present interesting results on the influence of Si addition to the technically important Fe-B alloys. The content of the work is in general well-explained and thoroughly discussed. There are some minor issues (see “detailed comments”) to be addressed before the paper can be accepted The English is in general good and needs just some minor revisions.

Detailed comments:

Page 2:

“and the number of M6C carbide increased rapidly”: Better: “and the amount of M6C carbide increased rapidly”

“… was listed in Table 1.” Should be: “… is listed in Table 1.”

“the preheated raw ferroboron were added into the furnace tafter deoxidize with a little pure aluminum.” I suppose you want to say: “the preheated raw ferroboron was added after deoxidizing it with a little pure aluminum.”

“And the definition diagram of longitudinal and transverse section in the DS Fe-B alloy as shown in Figures 1b and 1c, which means that the longitudinal section is parallel to the growth direction of Fe2B crystal, while the transverse section is perpendicular to the Fe2B growth direction [31,32].” This sentence is not comprehensible. Please rewrite.

Page 5:

“Obviously, the as-detected peaks”. Probably: “detected peaks” – unless you applied later some mathematical treatment to them.

“hard phase has more higher values than” Better: “hard phase has higher values than”

Page 9, Figure 9:

I have several problems with this figure:

1 1) Judging from the positions of the crosses in part (a) and the scale displayed in lower right edge it appears that the scan spans over a distance of at least 3.5 µm while the ordinate of part (b) covers only 2.5 µm.

2 2) The fit curve for A2 in part (b) reaches a negative value for Si addition close to 2 µm which does not make sense. Also, a deep minimum of this curve seems not be justified by any data points. You should look for a fit curve that runs rather flat between the two rightmost points which corresponds better to the expectations the reader has looking at the other two fits.

3   3) The ordinate description “spacing of boride/µm” is rather misleading. What I think you have here is the distance from the left edge of the channel you investigate with your scan. So “lateral distance/µm” or something like this should be more adequate.

 Some minor language issues:

·        *  Type errors on page 2: “crabide” instead of “carbide, “Frist” instead of “First”, “owe-way” instead of “one-way”

·        *  “Si element, Cr element” are not very elegant. Better just : “Si, Cr” or “silicon, chromium” if you like to vary at bit.

·         * There are many spaces missing after full stops, commas and parentheses.

Reviewer 3 Report

The article is well-structured and well-written: furthermore, the authors have done a good literature review to support their research. However, I consider that this research lacked an evaluation of the wear properties of these developed alloys, and the effect of Si on the wear properties of these alloys was not determined. On the other hand, it has been reported that steels with B have good behavior against wear. Therefore, I consider that this may be the weakness of this paper.

Reviewer 4 Report

Dear Authors,

Please find comments on the paper “Effect of Silicon Content on Microstructures and Properties of Directionally Solidified Fe-B Alloy”. The article describes the effects of Si content in the microstructure and properties of Fe-B alloys. The research performed in this work fits the scope of Materials, and it would interest the readers due to the significant contribution to the field of industrial wear-resistant materials. However, the manuscript requires some corrections to be published in this journal. Thus, minor corrections are suggested before the manuscript is considered for publication in Materials.

Suggestions:

Indicate why DS was used instead of conventional casting.

Indicate why corroded specimens were used for XRD. It is unclear why they were corroded and whether corrosion was made on purpose.

Please indicate the actual composition of the as-cast alloys and how it was performed. Indicate how the measurement of B was ensured by EDS analysis; once this has been done, include a table with average chemical compositions for Fe2B and α-Fe phases in the samples.

Include powder diffraction files (PDF cards) used for XRD phase identification in the materials and methods section.

Please check the spelling in section 3.2. Some sentences are difficult to understand.

Do not use “incline” please modify the sentence “Si is likely to incline to form the Si segregation at the ferrite/boride boundaries to refine the columnar structure of the DS Fe-B alloy.”

In Figures 6 and 7, it is possible to observe that Si partitions to one phase and B as well, but it is not clear where; mapping at higher magnification could help to elucidate this.

Figure 8 (a) does not indicate where the 5µm line scan for chemical analysis was taken because it seems that the length of the red arrow does not correspond in the image, and in 8 (b) add line scans data for Cr, Fe, B and in the same Si plot to compare (this could help to make things clear the information of mapping in figures 6 and 7.

It is not necessary to show the microsegregation of Si in the ferrite matrix in figure 9 as it is shown. Borides spacing changes along the grains, which is not constant; adding point analyses were taken in diagonal and not completely transversal in cell spacing. Besides, the ferrite matrix volume fraction increases as Si content increases in the alloy. The plot in Figure 9 b does not show that Si distribution in the matrix follows a parabolic law. The plot shows the microsegregation of Si in the matrix. However, the plot is not representative of all the alloys since cell spacing is not the same for all the alloys due to the Si content that enhances the ferrite matrix content and refines de Fe2B phase. A table with ferrite phase compositions for each alloy, giving the range of Si values in the phase, would show segregation instead of figure 9.

The authors are encouraged to emphasize what is new compared with other works in the field in their discussion.
